DATA RELEASE

# Polyploid genome assembly of *Cardamine chenopodiifolia*

Aurélia Emonet[1,†], Mohamed Awad[1,†], Nikita Tikhomirov[1,†],
Maria Vasilarou[1], Miguel Pérez-Antón[1,‡], Xiangchao Gan[1,2],
Polina Yu. Novikova[1] and Angela Hay[1,*]

1 Max Planck Institute for Plant Breeding Research, Carl-von-Linné-Weg 10, 50829, Köln, Germany
2 State Key Laboratory for Crop Genetics and Germplasm Enhancement, Bioinformatics Center, Academy for Advanced Interdisciplinary Studies, Nanjing Agricultural University, 210095, Nanjing, China

## ABSTRACT

*Cardamine chenopodiifolia* is an amphicarpic plant in the Brassicaceae family. Plants develop two fruit types, one above and another below ground. This rare trait is associated with octoploidy in *C. chenopodiifolia*. The absence of genomic data for *C. chenopodiifolia* currently limits our understanding of the development and evolution of amphicarpy. Here, we produced a chromosome-scale assembly of the *C. chenopodiifolia* genome using high-fidelity long read sequencing with the Pacific Biosciences platform. We assembled 32 chromosomes and two organelle genomes with a total length of 597.2 Mb and an N50 of 18.8 Mb. Genome completeness was estimated at 99.8%. We observed structural variation among homeologous chromosomes, suggesting that *C. chenopodiifolia* originated via allopolyploidy, and phased the octoploid genome into four sub-genomes using orthogroup trees. This fully phased, chromosome-level genome assembly is an important resource to help investigate amphicarpy in *C. chenopodiifolia* and the origin of trait novelties by allopolyploidy.

**Submitted:** 19 October 2024

\* Corresponding author. E-mail: hay@mpipz.mpg.de

† Contributed equally.

‡ Current address: Plant Science Research Laboratory (LRSV), UMR5546 CNRS/University of Toulouse 3, 24 chemin de Borde Rouge, 31320 Auzeville Tolosane, France.

Preprint submitted at https://doi.org/10.1101/2024.01.24.576990

**Subjects** Genetics and Genomics, Botany, Plant Genetics

## INTRODUCTION

*Cardamine chenopodiifolia* Pers. (NCBI:txid3101730) is an annual flowering plant belonging to the Brassicaceae family and is native to and widespread in South America [1–4]. *C. chenopodiifolia* is amphicarpic, meaning that each plant bears fruit both above and below ground, with two very distinct modes of seed dispersal (Figure 1). Exploding seed pods are produced above ground and disperse their many small seeds by explosive coiling of the fruit valves. Another type of seed pod develops below ground. The few large seeds produced by each of these fruits are released underground. The flowers on the main shoot of *C. chenopodiifolia* are positively geotropic and immediately grow towards the soil. These reduced flowers self-pollinate while growing through the soil and develop fruits that produce seeds underground. In contrast, the axillary shoots of *C. chenopodiifolia* grow away from gravity and produce open flowers that can also self-pollinate and develop explosive fruits [5]. The unique biology of *C. chenopodiifolia* makes it an ideal species for studying the development and evolution of the unusual trait of amphicarpy.

Cardamine is one of the largest genera in the Brassicaceae, with more than 200 species [6, 7], among which 58% are described as polyploids [8]. Based on chromosome counts, *C. chenopodiifolia* was described as an octoploid almost 100 years ago [9]. Another octoploid species in this genus, *C. occulta*, was recently sequenced as a model for ruderal weeds [10].

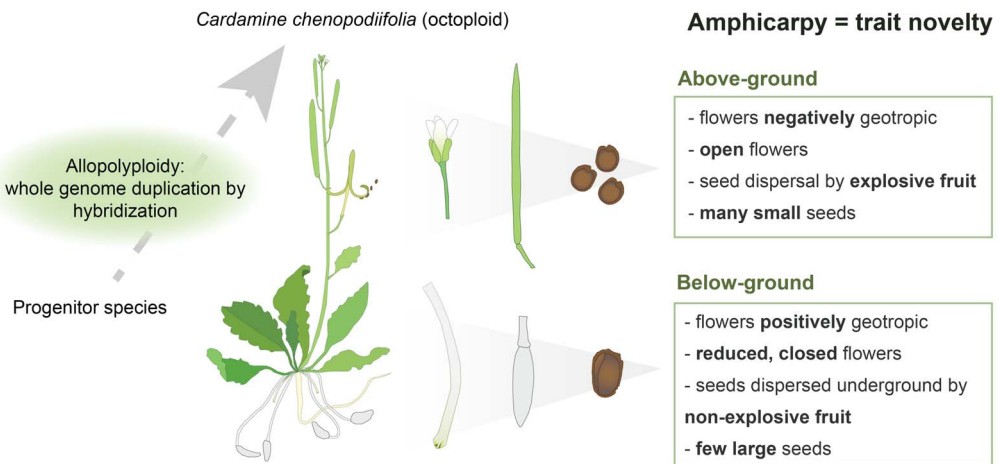

**Figure 1. Amphicarpy is associated with polyploidy in *Cardamine chenopodiifolia*.**
Amphicarpy is a rare trait where a single plant develops two different types of fruit – one above and another below the ground – with distinct dispersal strategies.

Polyploidy is a common feature of plant genomes and is believed to confer advantages for trait evolution [11]. Hence, the evolution of amphicarpy in *C. chenopodiifolia* may be linked to its octoploid genome. For this reason, assembling the genome of *C. chenopodiifolia* provides a useful resource for studying complex trait evolution by polyploidy.

Within *Cardamine*, the diploid species *C. hirsuta* is commonly used as an experimental system for comparative studies with its relative *Arabidopsis thaliana* [12–17]. The reference genomes and vast array of genetic tools in these two model plants make it advantageous to develop emerging model species within this phylogenetic neighborhood for comparative studies. This provides another motivation to assemble the genome of *C. chenopodiifolia* as a valuable tool for comparative studies and polyploidy research.

Here, we report the first whole-genome assembly for *C. chenopodiifolia* using single-molecule real-time sequencing technology from Pacific Biosciences (PacBio HiFi), and Omni-C technology. We assembled 32 chromosomes and phased the octoploid genome into four sub-genomes, each with eight chromosomes. *C. chenopodiifolia* likely originated via allopolyploidy with no clear evidence of biased fractionation among the four sub-genomes.

## RESULTS

### *C. chenopodiifolia* genome is octoploid

To verify the ploidy of the *C. chenopodiifolia* plants used for sequencing, we performed chromosome spreads using mitotic cells of flower buds and counted 64 chromosomes (Figure 2A). This suggests that *C. chenopodiifolia* has eight sets (octoploid) of eight chromosomes, typical of the ancestral cruciferrelative karyotype of $n = 8$ [18]. We then estimated its genome size by flow cytometry, comparing the nuclei DNA content of *C. chenopodiifolia* with *C. hirsuta* as a reference standard. *C. hirsuta* has a genome size of 198 Mb [19], and its nuclei were separated into five peaks representing DNA contents of 2C, 4C, 8C, 16C, and 32C (Figure 2D–E). Pooling nuclei from both species for flow cytometry allowed us to compare their relative DNA content. We identified three peaks belonging to *C. chenopodiifolia*, representing DNA contents of 2C′, 4C′, and 8C′ (Figure 2B–C). The first 2C′ peak of

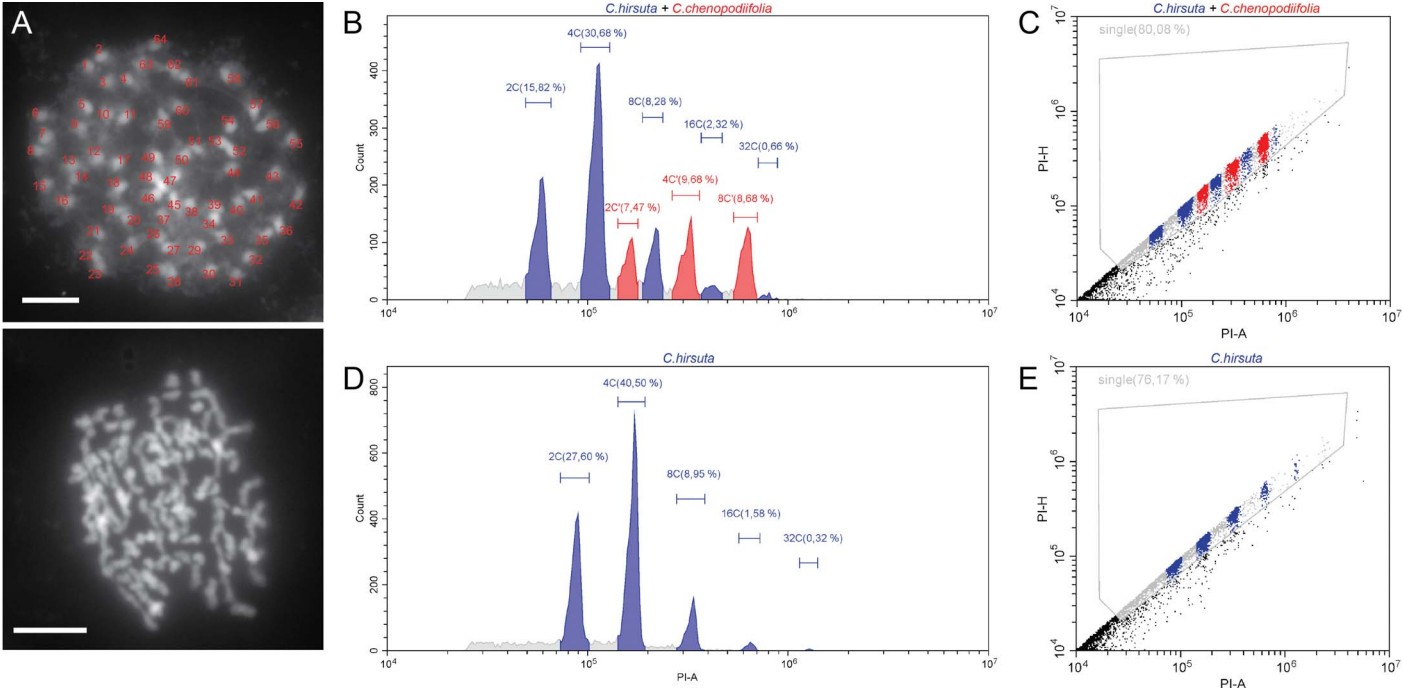

**Figure 2.** ***Cardamine chenopodiifolia* is octoploid and has 64 chromosomes.**
(A) Metaphase chromosome spreads in aerial *C. chenopodiifolia* flower cells stained with DAPI. Two examples are shown with chromosomes labeled 1–64 in the top image and unlabeled in the image below. (B–E) Flow cytometry analysis of a mixture of *C. hirsuta* and *C. chenopodiifolia* (B–C) or *C. hirsuta* alone (D–E). (B, D) The X-axis indicates the area of the signal from propidium iodide (PI-A) on a logarithmic scale. The Y-axis indicates the number of nuclei recorded (Count). Segments were drawn manually to cover each peak. Blue: *C. hirsuta*. Red: *C. chenopodiifolia*. The percentage of events corresponding to each peak over the total events is indicated. XC indicates the endopolyploidy level (e.g., 2C for nuclei in a diplophasic state). (C, E) Single nuclei events were discriminated and gated by assessing propidium iodide fluorescence area (PI-A) versus height (PI-H).

*C. chenopodiifolia* had a mean position (PI-area = 159,474.8) slightly lower than the 8C peak of *C. hirsuta* (Figure 2B,D). The relative position of these peaks is consistent with the expectation of an octoploid genome for *C. chenopodiifolia.* From this analysis, we estimated the genome size of *C. chenopodiifolia* to be 551 Mb (198 Mb × (159,474.8/57,315.6) – see Methods).

## *C. chenopodiifolia* chromosome-level genome assembly

To ensure homozygosity, *C. chenopodiifolia* plants were self-pollinated for five generations by single-seed descent using aerial seeds, before sequencing. High molecular-weight DNA was extracted from seedlings, and HiFi sequencing was performed using the PacBio Sequel II platform. We generated 103 Gb of raw DNA sequence (corresponding to 85× coverage) comprising 2,978,449 reads with a mean read length of 17.28 kb (Figure 3A, Table 3). Long reads were pre-assembled with HiCanu [20], Flye (RRID:SCR_017016), and Hifiasm (RRID:SCR_021069) software. We used these draft assemblies with GALA (Gap-free long-read assembly tool; RRID:SCR_026184) [21] to obtain a final chromosome-by-chromosome assembly without the need for Omni-C data. This resulted in an almost complete chromosome-level genome of 597 Mb with a scaffold N50 value of 18.8 Mb (Table 1). We obtained 32 chromosomes, one mitochondrial genome, and one plastid genome. Only one gap was left in the centromeric region of chromosome 9 (Table 1).



**Figure 3.** **Assembly of 32 gap-free chromosomes of *Cardamine chenopodiifolia*.**
(A) HiFi read density used for *C. chenopodiifolia* genome assembly. (B) Omni-C contact matrix showed no mis-assembly of the 32 chromosomes.

## Assessment of genome assembly quality

Genome-wide chromosome conformation capture methodologies, such as Hi-C and Omni-C, allow the chromosomal structure to be linked directly to genomic sequence and can,

**Table 1.** Statistics of *Cardamine chenopodiifolia* scaffolded genome assembly.

| Genome assembly | |
|---|---|
| Genome size (bp) | 597,266,257 |
| Chromosomes | 32 + 2 organelles |
| N50 (Mb) | 18.80 |
| N90 (Mb) | 15.87 |
| L50 | 15 |
| L90 | 28 |
| Longest Chromosome (Mb) | 24.07 |
| Gaps | 1 |
| GC (%) | 35.69 |
| Ploidy | Octoploid |

**Table 2.** HiFi read mapping and variant calling statistics for *Cardamine chenopodiifolia* genome assembly. The frequency of each feature is shown.

| Genomic feature | Genome assembly |
|---|---|
| Mapping statistics | |
| Mapped reads | 2,978,150 (99.98) |
| Unmapped reads | 299 ($1.00 \times 10^{-4}$) |
| Mismatches | 139,046,575 ($2.70 \times 10^{-3}$) |
| Insertions | 84,999,110 ($1.65 \times 10^{-3}$) |
| Deletions | 42,298,726 ($8.21 \times 10^{-4}$) |
| Indels | 127,297,836 ($2.47 \times 10^{-3}$) |
| Variant calling statistics | |
| Total variants | 20,800 |
| Single nucleotide polymorphisms (SNPs) | 140 |
| Indels | 20,660 |
| Multiallelic sites | 282 |
| Multiallelic SNPs | 0 |

therefore, be used to achieve chromosome-scale scaffolding. Since our initial assembly had already resolved the 32 chromosomes of *C. chenopodiifolia,* we used our Omni-C data to assess the assembly quality. We mapped 71,599,541 raw Omni-C reads to the genome assembly to generate a contact matrix. The resulting contact maps showed 32 unambiguous chromosomes with no obvious mis-assemblies (Figure 3B).

We also evaluated the completeness of the genome assembly using Benchmarking Universal Single-Copy Orthologs (BUSCO; RRID:SCR_015008). We identified 99.8% of the 1,614 conserved core Embryophyta genes as complete (18.9% as single-copy genes and 80.9% as duplicated genes) and only 0.1% of the genes as fragmented (Table 4).

When we mapped PacBio HiFi reads to the genome assembly with minimap2 (RRID:SCR_018550), we found the overall mapping rate was 99.98%, indicating that most of the sequencing data was represented in the assembly (Table 2). The distribution of sequence coverage depth was regular along all chromosomes except for some regions with lower coverage on chromosomes 4, 5, 14, and 32, and with higher coverage on chromosomes 9, 12, and 28, indicating extended and collapsed sequences, respectively (Figure 7).

To test the correctness of our assembly, we used BCFtools (RRID:SCR_005227) for variant calling (Table 2). A low number of variants is usually a sign of a correctly assembled genome. We obtained only 20,800 variants, most of them insertions or deletions, suggesting that our genome assembly is generally correct. This result is also indicative of a highly homozygous genome, as expected for a selfing species [5] that was subjected to repeated generations of selfing prior to sequencing (see Methods).



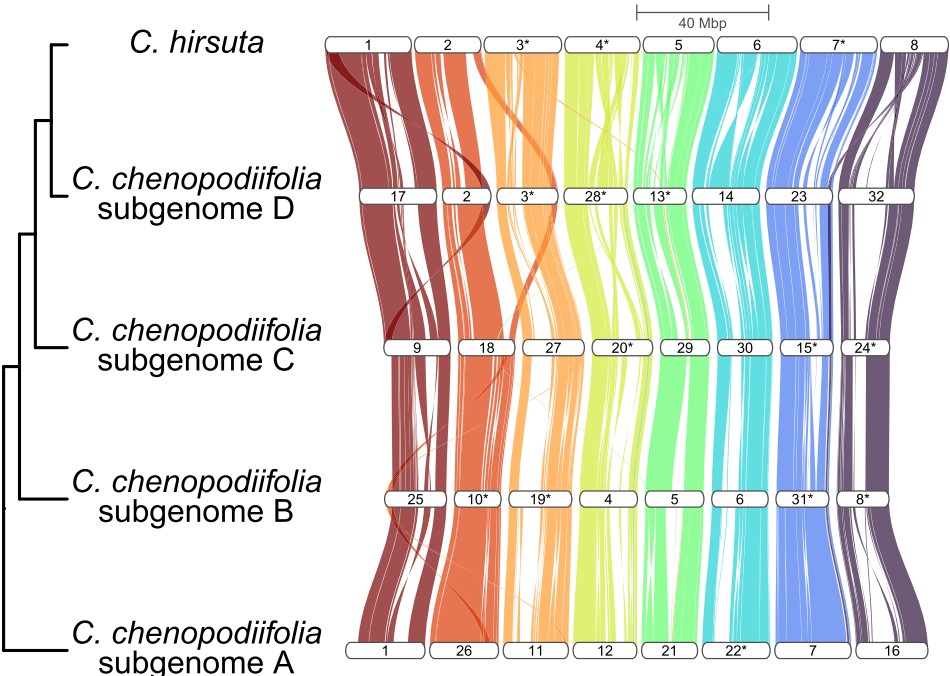

**Figure 4.** ***Cardamine chenopodiifolia* is allo-octoploid.**
GENESPACE riparian plot showing collinearity of the four sub-genomes of *C. chenopodiifolia* and the diploid *C. hirsuta* genome. Syntenic chromosomes are indicated by color. Asterisks (*) mark chromosomes that have been inverted for better visualization. The scale indicates sequence length in Mb.

Additionally, we used HISAT2 (RRID:SCR_015530) to align RNA-seq data from *C. chenopodiifolia* to the genome assembly. The transcriptome data comprised Illumina short reads and PacBio Iso-Seq long reads [5, 22]. Short reads aligned from 91.64% to 94.04%, while long reads aligned from 93.08% to 97.56%.

## The allo-octoploid genome of *C. chenopodiifolia* comprises four sub-genomes

Polyploids are typically classified as autopolyploids or allopolyploids, depending on their evolutionary history. Autopolyploids arise by whole genome duplication within a single species. In contrast, allopolyploids are the result of a hybridization event between different species [23, 24]. To investigate ploidy and distinguish between these two scenarios in *C. chenopodiifolia,* we assessed the large-scale similarity of chromosomes by synteny analysis. We observed co-linearity between the four sub-genomes of octoploid *C. chenopodiifolia* and the diploid *C. hirsuta* genome (Figure 4). We also observed profound structural variations among all homeologous chromosomes (Figure 4). This indicates that different homeologs are unlikely to recombine and thus supports an allopolyploid origin for *C. chenopodiifolia.*

In order to phase the 32 *C. chenopodiifolia* chromosomes into sub-genomes, we aimed to find the most frequently observed relationships between homeologous chromosomes. To this end, we generated rooted orthogroup trees using the proteomes of *C. chenopodiifolia,* the diploid species *C. hirsuta* [19], the octoploid species *C. occulta* [10], and an outgroup

*Crucihimalaya himalaica* [25]. We summarized these trees for each of the eight homeolog groups, which allowed us to resolve the relationships between their members and thus phase the four sub-genomes of *C. chenopodiifolia* (Figure 5).

To investigate whether the allopolyploid sub-genomes of *C. chenopodiifolia* exhibit obvious signs of biased fractionation, we compared gene density and GC content between the four sub-genomes (Figure 6). We observed no substantial differences in chromosome length or gene density between sub-genomes A, B, C, and D (Figure 6). Also, the pattern of GC content was overall similar between the four sub-genomes, giving no indication, for example, of biased accumulation of transposable elements (Figure 6). Therefore, we found no clear evidence for asymmetric sub-genome evolution in *C. chenopodiifolia*. In conclusion, the octoploid *C. chenopodiifolia* genome is composed of four different sub-genomes. We could confidently phase them into eight groups of four homeologous chromosomes, with a one-to-one correspondence to chromosomes 1 to 8 of *C. hirsuta*.

## DISCUSSION

Trait diversity is an important resource in light of climate change and biodiversity challenges. Investigation of trait diversity requires the characterization of unconventional model organisms and the establishment of genomic tools in these species. The Brassicaceae family is rich in species characterized by very diverse traits and different ecological adaptations [26]. Moreover, this family contains the model species *Arabidopsis thaliana* and many other species with sequenced genomes, thus providing rich genomic resources for comparative studies. This motivated us to develop the polyploid species *Cardamine chenopodiifolia* as an emerging model organism in the Brassicaceae to study the unusual trait of amphicarpy [5].

Here, we provide a chromosome-level genome assembly for *C. chenopodiifolia*. Using the PacBio platform, we generated a 597.2 Mb octoploid genome assembly composed of 32 chromosomes ($2n = 8\times = 64$), with an N50 length of 18.8 Mb. Both the estimated ploidy and genome size of this assembly agree well with our results from cytology and flow cytometry. Differences between the four sub-genomes of *C. chenopodiifolia*, coupled with low heterozygosity in this selfing plant, may have facilitated the chromosome-level haploid assembly that we produced without the need for Omni-C scaffolding.

We used this *C. chenopodiifolia* genome assembly to investigate its polyploid origin. Octoploids are usually expected to arise from the whole genome duplication of a tetraploid, or by the hybridization of two different auto- or allotetraploid species. These scenarios are considered more likely due to the challenges inherent to successful meiosis in a newly formed polyploid [27]. Our synteny analysis revealed many rearrangements between *C. chenopodiifolia* homeologous chromosomes, which is an unexpected scenario for an autopolyploid with random chromosome pairing. In addition, there was no clear pattern of pairwise similarity, as expected from a merger of two divergent autotetraploid genomes. Therefore, the four dissimilar sub-genomes of *C. chenopodiifolia* might have resulted, for example, from the merger of two allotetraploids with non-overlapping parental species. One such example of a *Cardamine* allotetraploid species is *C. flexuosa*, which is a natural hybrid between *C. amara* and *C. hirsuta* [28]. Since polyploids make up more than half of the large number of species in *Cardamine* [7, 8], extant allotetraploids may exist that descended from the possible progenitors of *C. chenopodiifolia*.

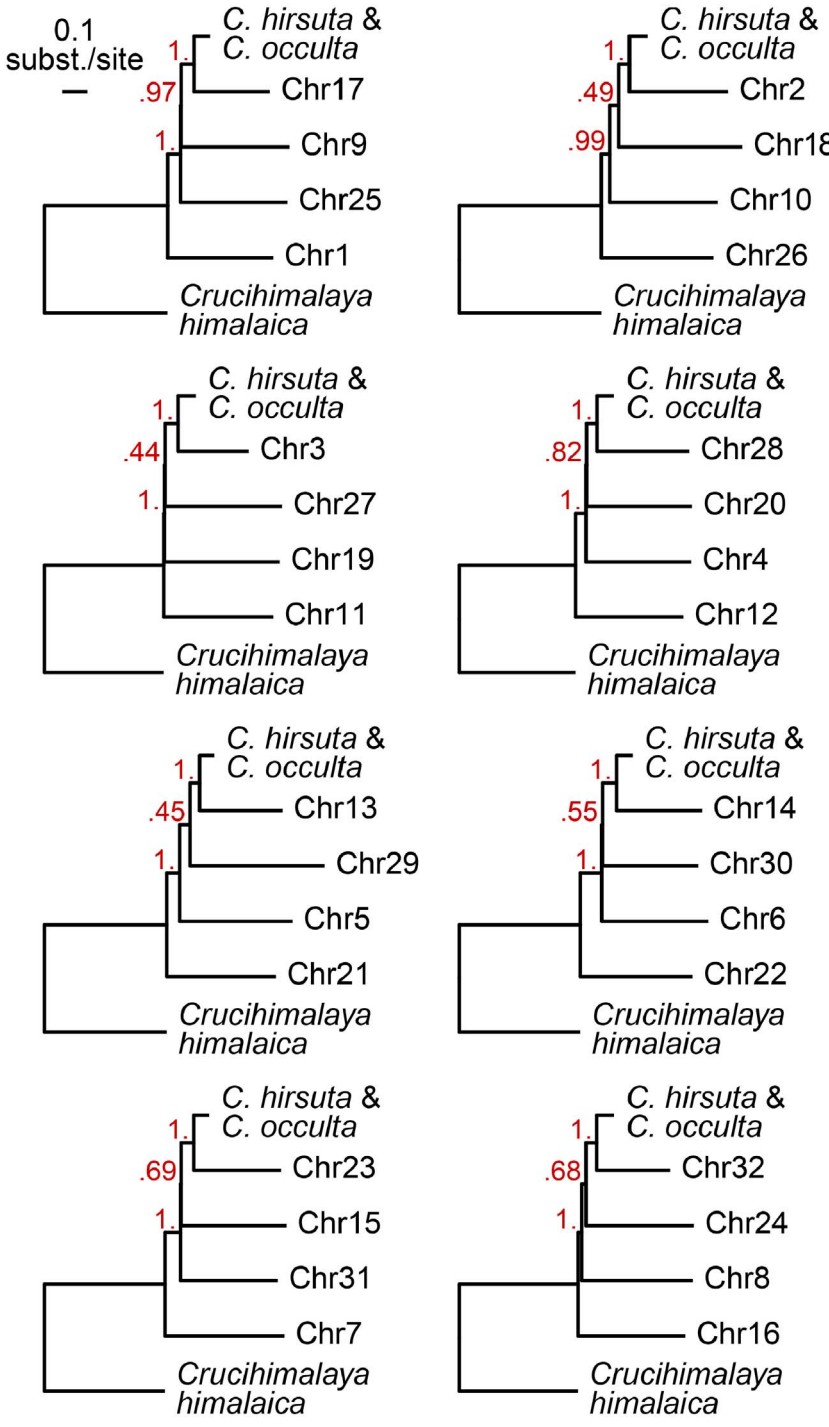

**Figure 5.** **Phasing the octoploid *Cardamine chenopodiifolia* genome into four sub-genomes.**
Summary of orthogroup trees for each of the eight homology groups of *C. chenopodiifolia* chromosomes. The relationship between homeologous chromosomes in each group phased the 32 chromosomes into four sub-genomes.

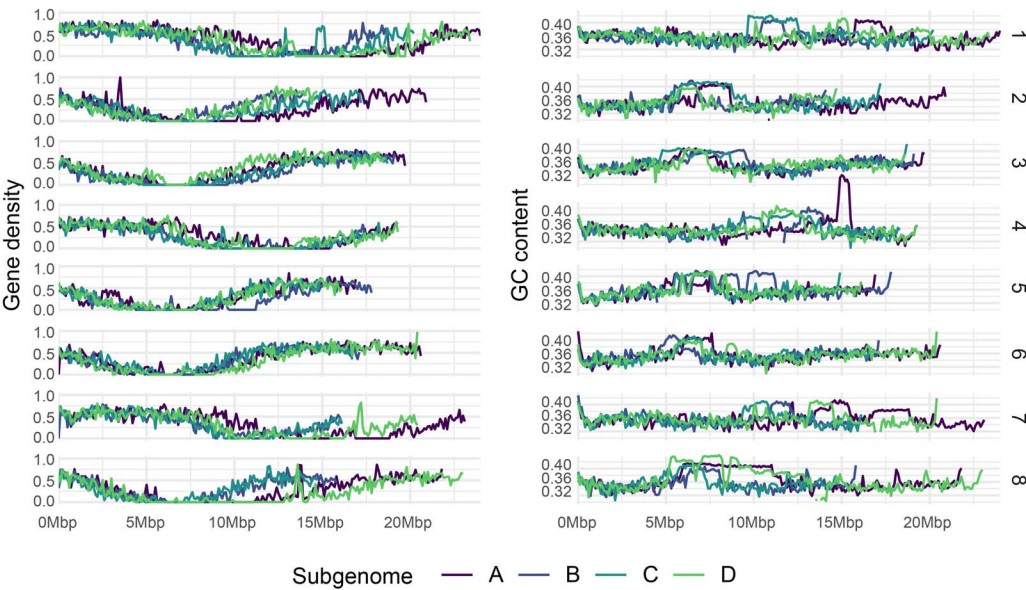

**Figure 6. Comparisons of gene density and GC content between the four sub-genomes of *Cardamine chenopodiifolia*.**
Plots of gene density and GC content along the chromosomes are compared between sub-genomes A (indigo), B (blue), C (cerulean), and D (green) of *C. chenopodiifolia*. Chromosome groups are labelled 1–8 to the right of the plots. Some chromosomes were inverted for better visualization, as indicated in Figure 4. Total length of sub-genome A: 166,775,643 bp; B: 136,630,202 bp; C: 137,163,421 bp; D: 156,164,804 bp.

Given that the progenitors of *C. chenopodiifolia* are currently unknown, or perhaps even extinct, we could not use this information to separate the sub-genomes as was done, for example, in peanut [29, 30] or wheat [31]. Therefore, we used available *Cardamine* genomes to construct orthogroup trees and phased the sub-genomes of *C. chenopodiifolia* on the basis of the most commonly observed relationships between chromosomes. Sub-genome phasing is an important step towards understanding the evolution and adaptive potential of *C. chenopodiifolia*, and in particular, to elucidate the origin of amphicarpy in this species.

Polyploidy is thought to be a mechanism by which plants can adapt to stressful conditions, providing increased genetic variation and a better chance to evolve beneficial adaptations [11, 25]. Amphicarpic species are reported to grow in disturbed habitats [32], raising the possibility that polyploidy might have contributed to the *de novo* evolution of amphicarpy in *C. chenopodiifolia* as an adaptation to disturbed environments. Alternatively, geocarpy might have been a trait inherited from a progenitor species, contributing to the evolution of amphicarpy as a bet-hedging strategy for seed dispersal in variable conditions. Although geocarpy has not been reported in *Cardamine*, this trait is present in the more distantly related species *Geococcus pusillus* J. Drum. in the Brassicaceae family [32]. Whether undescribed or extinct geocarpic species in *Cardamine* contributed to the origin of *C. chenopodiifolia* is an interesting question to be addressed in future studies. In this regard, the *C. chenopodiifolia* genome is a valuable genomic resource for studying the development and evolution of amphicarpy by allopolyploidy.



## MATERIAL AND METHODS

### Growth conditions

*Cardamine chenopodiifolia* seeds (Ipen: XX-0-MJG-19—35600) were obtained from the Botanic Garden of the Johannes Gutenberg University, Mainz, Germany. The country of origin of the original collection was undocumented. Plants were propagated for five generations by single seed descent from self-pollinated aerial flowers to ensure homozygosity. Aerial seeds were germinated in long-day conditions on 1/2 Murashige and Skoog plates after 7 days of stratification. Then, 1-week-old seedlings were transferred to soil and grown in a walk-in chamber (16 h light, 20 °C; 8 h dark, 18 °C; 65% humidity). *Cardamine hirsuta*, herbarium specimen voucher Hay 1 (OXF) [13], was cultivated on soil in long-day conditions (LD; days: 20 °C, 16 h; nights: 18 °C, 8 h) after stratification on soil at 4 °C in the dark for 7 days.

### Chromosome spreads

Mitotic chromosome spreads were performed as previously described [33] with minor modifications. Inflorescences were immediately fixed in fresh 3:1 Clarke's fixative (3 vol. absolute ethanol: 1 vol. acetic acid). Fixative was refreshed three times. After fixation, inflorescences were dissected under a binocular, and white closed buds were collected (no pollen present). Samples were washed twice for 2 min in deionized water, and twice for 2 min in 10 mM trisodium-citrate buffer (pH 4.5, adjusted with HCl). Samples were digested for 1 hour and 45 min at 37 °C (digestion mix: 0.3% (w/v) Pectolyase Y-23 (MP Biomedicals), 0.3% (w/v) Driselase (Sigma), 0.3% (w/v) Cellulase Onozuka R10 (Duchefa), 0.1% sodium azide in 10 mM tri-sodium-citrate buffer). Three to six buds were transferred to a clean slide in a drop of water and dilacerated with a thin needle until it formed a suspension. Next, 10 µl of 60% acetic acid was delicately incorporated into the suspension with a hooked needle, and then the slide was heated on a hot block at 45 °C for 1 min while slowly stirring. Another 10 µl of 60% acetic acid was added as the drop started to evaporate and stirred for a supplementary minute. To mount the slide, ice-cold fresh fixative solution was pipetted as a boundary around the droplet and allowed to invade the slide. Then, a jet of ice-cold fixative was applied twice directly onto the center of the circle. After the removal of excess fixative by tilting, the slide was dried at room temperature. Once dried, 8 µl of DAPI solution (2 µg/ml in antifade mounting medium Citifluor AF1, Agar Scientific) was added onto a coverslip and mounted on the slide. Imaging was performed using a Zeiss Axio Imager Z2 microscope and Zen Blue software. Images were acquired with a Plan-Apochromat 100×/1.40 Oil M27 objective, Optovar 1.25× Tubelens. The excitation and detection windows for DAPI were set as follows: excitation, 335–385 nm, detection, 420–470 nm.

### Flow cytometry

Flow cytometry was performed as previously described [34] with minor modifications. To release nuclei, newly expanded leaves of *C. hirsuta* and *C. chenopodiifolia* were chopped with a sharp razor blade on a petri dish containing 300 µL of Galbraith's buffer (45 mM $MgCl_2$, 20 mM MOPS, 30 mM sodium citrate, 0.1% (v/v) Triton X-100) [35] and including 50 mg/L RNAase. Nuclei suspensions were passed through 50 µm CellTrics® filters and stained with propidium iodide (PI) at a final concentration of 50 mg/L for 1 h on ice in darkness. Stained nuclei of *C. hirsuta* were analyzed separately and in combination with *C. chenopodiifolia* in a CytoFLEX (Beckmann Coulter) platform using the excitation and

**Table 3.** Statistics of the PacBio HiFi dataset for *Cardamine chenopodiifolia*.

| HiFi reads | |
| --- | --- |
| Total bases (Gb) | 103 |
| Total reads | 2,978,449 |
| Read N50 (kb) | 17.90 |
| Read mean (kb) | 17.28 |
| Read L50 | 1,225,263 |
| Coverage | 85× |

emission parameters for PI. We recorded 10,000 events for each sample, and gating was employed to exclude doublets and debris. Gating, analysis, and plotting were performed using the manufacturer's software (CytEXPERT).

Estimated genome size was calculated with the following formula, where *GS = genome size*, *2C = mean peak position* (PI-area):

$$\text{GS}_{C.\ chenopodiifolia} = \text{GS}_{C.\ hirsuta} \times \frac{2\text{C}_{C.\ chenopodiifolia}}{2\text{C}_{C.\ hirsuta}}$$

### DNA extraction, library construction, and sequencing

For genome sequencing, high-molecular-weight (HMW) DNA was isolated from 2 g fresh, shock-frozen seedlings (in liquid nitrogen) with the NucleoBond HMW DNA kit (Macherey Nagel, Düren, Germany) and DNA quality was assessed by capillary electrophoresis (Agilent FEMTOpulse). For PacBio library preparation, the HMW DNA was fragmented with g-tubes (Covaris) to get 20 kbp fragments, and then a library was prepared according to the recommendations of the SMRTbell Express Template Prep Kit 2.0 (Pacific Biosciences). Next, a size selection was applied to enrich for ≥10 kbp fragments (BluePippin, Sage Sciences), followed by long-read sequencing on four SMRT cells on a Sequel II device with Sequel II Binding kit 2.0, Sequel II SMRT 8M cells, and Sequel II Sequencing Plate 2.0 chemistry for 30 hours and a final concentration of 110 pmolar on plate. Sequencing was performed at the Max Planck Genome-centre Cologne. In parallel, a chromatin-capture library (Omni-C, Dovetail) was prepared according to recommendations from the vendor, followed by 2 × 150 paired-end sequencing on an Illumina NextSeq 2000 device at Max Planck Genome Centre Cologne resulting in 71,599,541 reads.

### Assembly of *C. chenopodiifolia* genome

We used GALA (gap-free long-read assembly tool, version 1.0.0) for *de novo* assembly of the *C. chenopodiifolia* genome [20]. Since GALA uses preliminary assemblies to cluster long reads into multiple groups for chromosome-by-chromosome data analyses, three draft assemblies were constructed from 85× coverage PacBio Hifi reads (Table 3) using HiCanu v2.1 [20], Flye v2.4 (RRID:SCR_017016) [36] and Hifiasm 0.5-dirty-r247 (RRID:SCR_021069) [37]. Assembly was conducted using default parameters and an expected genome size of 600 Mb.

The straightforward application of GALA generated 35 scaffolding groups. Among them, two scaffolding groups were assembled into single contigs with telomeric motifs at one end, indicating that each group represented a chromosome arm. We thus merged these two groups into a single scaffolding group and performed single-chromosome assembly using the LGAM module of GALA. The assembly of GALA was gap-free and complete, containing 32 pseudomolecules and two organelle chromosomes.



**Table 4.** Benchmarking Universal Single-Copy Orthologs (BUSCO) in *Cardamine chenopodiifolia* genome assembly.

| BUSCO Category | Value | Percent (%) |
| --- | --- | --- |
| Complete | 1,611 | 99.8 |
| Single copy | 305 | 18.9 |
| Duplicated | 1,306 | 80.9 |
| Fragmented | 1 | 0.1 |
| Missing | 2 | 0.1 |

We then polished the GALA assembly to enhance the assembly's correctness. The HiFi raw reads were mapped to the GALA assembly using minimap2 v. 2.17-r941 and the command 'minimap2 -x asm20' [38]. Then, we used an in-house genome polisher to enhance the correctness of the assembled genome [39].

## Assembly quality validation

Assembly contiguity was assessed with a Python script (see code availability). Assembly completeness was assessed by Benchmarking Universal Single-Copy Orthologs (BUSCO v.5.4.4) (RRID:SCR_015008) [40] with the dataset Embryophyta_odb10 (Table 4).

To assess correctness, we used 'minimap2 -x asm20' to map the HiFi reads to the final assembly. Then, we collected the mapping statistics from samtools-stats (RRID:SCR_002105) [41, 42]. Finally, we called the variants and collected the variant calling statistics using BCFtools [43]. To evaluate collapsing, we mapped the reads using minimap2 v2.20 with the '-x map-hifi' preset. Then, we collected the depth information using samtools-depth and marked all the regions with depth = average depth × 2 as collapsed regions. Average sequencing depth is shown in 5 kbp windows (Figure 7). In addition, we analyzed GC content and gene density (the latter obtained from the draft annotation made with Helixer [44]) using bedtools v2.30 (RRID:SCR_006646). The Omni-C data were analyzed with Juicer v1.6 (RRID:SCR_017226) [45], and the contact map was visualized in Juicebox (RRID:SCR_021172) [46].

## Alignment to *C. chenopodiifolia* transcriptome

*C. chenopodiifolia* long and short read transcripts (obtained from [5, 21]) were aligned to the *C. chenopodiifolia* assembly using HISAT2 v2.1.0 [47].

## Synteny analysis

We assessed the large-scale similarity of chromosomes in the assembly using the GENESPACE v1.3.1 pipeline [48] with a draft annotation obtained from the Helixer web interface [44]. Prior to the analysis, we manually removed a gene prediction error in a repetitive region of Chr7 (gene models ccheno_Chr7_002034.1 – ccheno_Chr7_006412.1) where numerous adjacent similar-sized genes were predicted. We made the required protein-sequence files using the AGAT v1.4.0 GFF parser (agat_sp_extract_sequences.pl -p) [49]. We visualized the results using R v4.2.2 with the tidyverse v2.0.0 (RRID:SCR_019186) data manipulation suite [50]. Some repetitive tasks were sped up with GNU parallel [51].

## Sub-genome phasing

We inferred the rooted orthogroup trees for *Cardamine chenopodiifolia, C. hirsuta* [19], *C. occulta* [10], and *Crucihimalaya himalaica* [25] proteomes using OrthoFinder v2.5.5



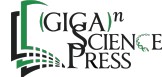

**Figure 7.** **Depth of sequence coverage along *Cardamine chenopodiifolia* chromosomes 1 to 32.**
Sequence coverage was calculated with minimap2 (blue lines); average depth (dashed black lines) and twice the average depth (dashed orange lines) are indicated.

(RRID:SCR_017118) [52]. We downloaded the genome annotations and proteome sequences for *C. hirsuta* from [53] and for *Crucihimalaya himalaica* from [54]. As the genome annotation of *C. occulta* was not available in public databases, we made a draft annotation using the Helixer web interface [44].

We grouped the obtained orthogroup trees for each of the eight sets of homologous chromosomes using Newick Utilities v1.1.0 [55] and summarized them for each set using ASTRAL-Pro3 [56]. When grouping the full set of orthogroup trees for each set of homologous chromosomes, the trees were modified to contain only leaves belonging to

these chromosomes using nw_rename, and clades with all members belonging to the same organism were collapsed using nw_condense.

## AVAILABILITY OF SOURCE CODE

The script for assessing assembly contiguity is available at [58]. The source code of GALA (RRID:SCR_026184) is available from GitHub at [59] under the MIT license.

## DATA AVAILABILITY

The PacBio and Omni-C raw read data and the genome assemblies generated by GALA in this study have been deposited at the European Nucleotide Archive (ENA) PRJEB71776. Additional supporting data is available in GigaDB [57].

## ABBREVIATIONS

HMW, high-molecular-weight; PI, propidium iodide; SNP, single nucleotide polymorphism.

## DECLARATIONS

### Ethics approval

The authors declare that ethical approval was not required for this type of research.

### Competing interests

The authors declare no competing interests.

### Authors' contributions

Conceptualization, AE, MA and AH; Investigation, AE, MA, NT, MV and MPA; Writing, AE, NT and AH; Funding Acquisition, AE, PYN and AH; Supervision, PYN, XG and AH.

### Funding

This work was supported by Swiss National Science Foundation fellowship P500PB_203021 to AE and Deutsche Forschungsgemeinschaft (DFG) grants – project numbers 462181533 and 497665889 – to PYN.

### Acknowledgements

We thank R. Mercier and A. Kalde for helping perform chromosome spreads, B. Huettel for PacBio sequencing, and W. Faigl for technical assistance. AH gratefully acknowledges support from a Max Planck Society core grant to the Department of Comparative Development and Genetics.

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
