## [Editor Report]

Editor’s AssessmentThis work presents the genome of Cardamine chenopodiifolia, an amphicarpic plant (developing two fruit types, one above and another below ground) in the mustard (Brassicaceae) family. Cardamines also known as bittercresses and toothworts. As an octoploid species it has been challenging to create a genome reference for this species, and in this case the authors finally managed to achieve this using PacBio HiFi long-reads and Omni-C technology to assemble a fully phased, chromosome-level genome. Obtaining a 597Mb genome assembled into 32 phased chromosomes (plus mitochondrial and plastid genomes), and only having one gap in the centromeric region of chromosome 9. Peer review asked for additional QC and benchmarking, helping demonstrate the genome quality was very high, with only one gap and a N50 of 18.80Mb. The data presented here potentially helping to develop this species as an emerging model organism in the Brassicaceae for studying the development and evolution of amphicarpy by allopolyploidy.Editor’s AssessmentThis work presents the genome of Cardamine chenopodiifolia, an amphicarpic plant (developing two fruit types, one above and another below ground) in the mustard (Brassicaceae) family. Cardamines also known as bittercresses and toothworts. As an octoploid species it has been challenging to create a genome reference for this species, and in this case the authors finally managed to achieve this using PacBio HiFi long-reads and Omni-C technology to assemble a fully phased, chromosome-level genome. Obtaining a 597Mb genome assembled into 32 phased chromosomes (plus mitochondrial and plastid genomes), and only having one gap in the centromeric region of chromosome 9. Peer review asked for additional QC and benchmarking, helping demonstrate the genome quality was very high, with only one gap and a N50 of 18.80Mb. The data presented here potentially helping to develop this species as an emerging model organism in the Brassicaceae for studying the development and evolution of amphicarpy by allopolyploidy.

---

## [Reviewer Report]

Reviewer name and names of any other individual's who aided in reviewer Rie ShimizuDo you understand and agree to our policy of having open and named reviews, and having your review included with the published papers. (If no, please inform the editor that you cannot review this manuscript.)YesIs the language of sufficient quality?YesPlease add additional comments on language quality to clarify if needed
Are all data available and do they match the descriptions in the paper? YesAdditional CommentsAre the data and metadata consistent with relevant minimum information or reporting standards? See GigaDB checklists for examples <a href="http://gigadb.org/site/guide" target="_blank">http://gigadb.org/site/guide</a>YesAdditional CommentsIs the data acquisition clear, complete and methodologically sound?YesAdditional CommentsIs there sufficient detail in the methods and data-processing steps to allow reproduction?YesAdditional CommentsIs there sufficient data validation and statistical analyses of data quality? YesAdditional CommentsIs the validation suitable for this type of data?YesAdditional CommentsIs there sufficient information for others to reuse this dataset or integrate it with other data?YesAdditional CommentsAny Additional Overall Comments to the AuthorThis manuscript deciphers the complicated genome of an octoploid species, Cardamine chenopodiifolia. They successfully assembled a chromosome-level genome with 32 chromosomes, consistent with the chromosome counting. They evaluated the quality of the genome by several methods (mapping Omni-C reads, BUSCO, variant calling etc.). All benchmarks ensured the high quality of their assembly. They even tried to phase the chromosomes into four subgenomes, and one subgenome was successfully phased thanks to its higher divergence compared to the other three sets. Despite their intensive effort, the other three subgenomes could not be phased, suggesting the relationship originated from the same or closely related species. As a whole, the manuscript is very well written and describes enough details, and the genome data looks like it is already available in a public database. They even added a description of the biological application of this assembly about the amphicarpy. I only found a few minor points for which I kindly suggest reconsideration/rephrasing before publication, as listed below.
*As the review PDF does not contain the line numbers, I suggest the original description at the first line and then write my comments. -C. chenopodiifolia genome is octoploid …, suggesting that its genome is octoploid. They compare the 8C peak of C. hirsuta and 2C peak of the target, but considering the genome size variation among Cardamine species, I do not think this is an appropriate expression. The pattern may mean ‘consistent’ with the expectation from C. hirsuta peaks but does not ‘suggest’ octoploidy. -C. chenopodiifolia chromosome-level genome assembly PacBio Sequel II platform. Here and nowhere, they do not mention the mode of sequencing (only found in method and the title of a table). Maybe ‘HiFi’ could be added here to make the method clearer. -Table 2. It would make more sense to overview the genome quality if the N90 and L90 (or similar, if it is already fragmented at L90) values are added. (maybe the same for Table 1). Otherwise Nx curves would be also fine for the same purpose. -We obtained only 20800 variants,…as expected for a selfing species. It might be partially due to selfing in wild habitat, but also by selfing (5 times) in the lab. This should be mentioned here to avoid misleading. -Table 4 The unit of each item (bp, number, frequency…?) should be suggested. In addition to the points listed above, I appreciate more Information about the phased chromosomes set: Total subgenome sizes of this set and the other three sets?(1:3 or imbalanced?) It would be even better with a synteny plot in addition to the colinear plot as Fig 3C. (e.g. by GENESPACE or something similar, including phased and unphased chenopodiifolia chromosome sets and C. hirsuta)RecommendationMinor Revision

---

## [Reviewer Report]

Reviewer name and names of any other individual's who aided in reviewer Qing LiuDo you understand and agree to our policy of having open and named reviews, and having your review included with the published papers. (If no, please inform the editor that you cannot review this manuscript.)YesIs the language of sufficient quality?YesPlease add additional comments on language quality to clarify if needed
Are all data available and do they match the descriptions in the paper? YesAdditional CommentsAre the data and metadata consistent with relevant minimum information or reporting standards? See GigaDB checklists for examples <a href="http://gigadb.org/site/guide" target="_blank">http://gigadb.org/site/guide</a>YesAdditional CommentsIs the data acquisition clear, complete and methodologically sound?YesAdditional CommentsIs there sufficient detail in the methods and data-processing steps to allow reproduction?YesAdditional CommentsIs there sufficient data validation and statistical analyses of data quality? YesAdditional CommentsIs the validation suitable for this type of data?YesAdditional CommentsIs there sufficient information for others to reuse this dataset or integrate it with other data?YesAdditional CommentsAny Additional Overall Comments to the AuthorThis manuscript “Polyploid genome assembly of Cardamine chenopodiifolia” produced a chromosome-scale assembly of the octoploid C. chenopodiifolia genome using highfidelity long read sequencing with the Pacific Biosciences platform with two organelle genomes with a total length of 597.2 Mb and an N50 of 18.8 Mb together with BUSCO analysis (99.8% genome completeness), and phased one of the four sub-genomes. This study provides a valuable resource to investigate the understudied trait of amphicarpy and the origin of new traits by allopolyploidy. The manuscript is suitably edited and significant data for amphicarpy breeding of C. chenopodiifolia except for the below revision points. The major revision is suggested for the current version of the manuscript. 1 Please elucidate “an N50 of 18.8 Mb”, which is Contig or Scaffold N50 length. 2 Please elucidate “originated via allo- rather than auto-polyploidy”, which is “originated via allopolyploidy rather than autopolyploidy”. 3 Please substitute the word “understudied trait” using alternative sensible word. 4 “to phase this set of chromosomes by gene tree topology analysis”, it is suggested to be “to phase this set of chromosomes by gene phylogeney analysis”. 5 In the first section of Resuts, Cardamine chenopodiifolia genome is octoploid is suggested. 6 Could Table 1 and Table2 be combined as one table to present the sequencing and assembly characterization of C. chenopodiifolia genome. 7 Could the entromere locations be predicted in Table 5, which is the 32 chromosome summary of C. chenopodiifolia genome. 8 In Table 2, assembly 32 chromosomes including two organelles, which is not close related with the C. chenopodiifolia genome, from my point of view, two organelle genome assembly do not critical section of manuscript. 9 Could all figure numbers are ordered below each group figures, for example the below figure should be numbered before the Figure 2A (according group figure presence order). I wonder it is Figure 2, authors want to elucidate the chromosome number 2n=42, while I can’t count out 42 chromosomes from present format.Could authors using alternative clear figure to show the cytological evidence of C. chenopodiifolia chromosome number. 10 In Figure 5A, it is difficult to point out the clear meaning for first-diverged chromosome from gene tree, which is a phylogenetic meaning tree or just framework, could author redraw this Figure 5A in order to reader got what you mean.RecommendationMajor Revision

---

## [Reviewer Report]

Reviewer name and names of any other individual's who aided in reviewer Kang ZhangDo you understand and agree to our policy of having open and named reviews, and having your review included with the published papers. (If no, please inform the editor that you cannot review this manuscript.)YesIs the language of sufficient quality?YesPlease add additional comments on language quality to clarify if needed
Are all data available and do they match the descriptions in the paper? YesAdditional CommentsAre the data and metadata consistent with relevant minimum information or reporting standards? See GigaDB checklists for examples <a href="http://gigadb.org/site/guide" target="_blank">http://gigadb.org/site/guide</a>YesAdditional CommentsIs the data acquisition clear, complete and methodologically sound?YesAdditional CommentsIs there sufficient detail in the methods and data-processing steps to allow reproduction?YesAdditional CommentsIs there sufficient data validation and statistical analyses of data quality? YesAdditional CommentsIs the validation suitable for this type of data?YesAdditional CommentsIs there sufficient information for others to reuse this dataset or integrate it with other data?YesAdditional CommentsAny Additional Overall Comments to the AuthorThe paper produced a chromosome-scale assembly of the C. chenopodiifolia genome in the Brassicaceae family, and offers a valuable resource to investigate the understudied trait of amphicarpy and the origin of new traits by allopolyploidy. I have the following comments which can be considered to improve the ms. Major points. 1.The introduction states that Cardamine is among the largest genera within the Brassicaceae family. The octaploid model species C. occulta and the diploid C. hirsuta have been sequenced. Therefore, I propose that a description of the evolutionary relationships among various species be included here. Additionally, the significance of the amphicarpic trait in the study of plant evolution and adaptation could be highlighted when discussing their octoploid characteristics. 2.The paper omits a detailed description of genome annotation and significant genomic features, which are essential for clearly illustrating the characteristics of the genome. To enhance this aspect, it would be beneficial to include a circular chart that displays fundamental components such as gene density, CG content, TE density, and collinearity links, among others. 3.The authors employed various techniques to differentiate the four subgenomic sets within the C. chenopodiifolia genome and ultimately managed to isolate a single sub-genomic set. The paper references the assembly of the octaploid genome of another model plant, C. occulta, within the same genus. Could it be utilized to compare with C. chenopodiifolia to achieve improvements? In addition, I suggest the authors to examine the gene density differences among these subgenomes, which could be helpful in distinguishing them. 4.Little important information were included in Table 1, 3, and Figure 4. These tables and figures should be moved to Supplementary data. 5.Evidence from Hi-C heatmap should be provided to validate the structural variations among different sets of subgenomes, such as those in Figure 3. Minor points. 1.Figure 5B, please change the vertical coordinate ‘# gene pairs’ to ‘Number of gene pairs’. The fonts in some figures are a little bit small. I suggest to adjust them to make it easy to read.RecommendationMinor Revision